# Qualitative Analysis of a Home-Delivered Produce Prescription Intervention to Improve Food and Nutrition Security

**DOI:** 10.3390/nu16234010

**Published:** 2024-11-23

**Authors:** Graciela Caraballo, Hemen Muleta, Anar Parmar, Noah Kim, Qadira Ali, Laura Fischer, Kofi Essel

**Affiliations:** 1School of Medicine and Health Sciences, George Washington University, Washington, DC 20052, USA; gcarab@stanford.edu (G.C.);; 2Department of Obstetrics and Gynecology, Stanford University School of Medicine, Stanford, CA 94305, USA; 3Children’s National Hospital, Washington, DC 20010, USA; hmuleta@montefiore.org (H.M.);; 4Department of Pediatrics, Albert Einstein College of Medicine, Bronx, NY 10461, USA; 5Elevance Health, Indianapolis, IN 46204, USA

**Keywords:** food insecurity, pediatric obesity, diet-related chronic diseases, produce prescription, nutrition security, community clinical collaboration

## Abstract

**Background/Objective:** In total, 17.9% of households with children experienced food insecurity (FI) in 2023. Produce prescription interventions (PRx) are a viable intervention to address FI and improve diet quality. Few studies have explored home-delivered PRxs in children. The objective of this qualitative study is to explore the experience of a novel PRx among families with young children in households at risk of experiencing FI and diet-related chronic disease. **Methods:** Semi-structured interviews were conducted with caretakers after the completion of a 12-month PRx. Interviews were recorded, transcribed, and analyzed using thematic analysis to identify emergent themes. Univariate descriptive statistics were used to describe baseline demographics. **Results:** Twenty-five families were enrolled, from which eighteen completed the program and fifteen agreed to participate in an interview. All participants were African American women. The mean age was 30.2 (±6.4) years old, and the median household size was three. Qualitative data analysis revealed three major themes. (1) The produce delivery partially alleviated financial stress, contributing to increased produce consumption patterns; (2) the intervention positively shifted the nutrition- and cooking-related knowledge and behavior of families; and (3) familial and programmatic barriers affected participation and engagement. **Conclusions:** PRxs are a viable option to support families to lessen the burden of FI from financial hardship and build healthy dietary habits. These insights can inform future PRx program development, delivery, evaluation, and policy or funding decisions. Future research should examine the sustained impact of PRx on healthy eating, health outcomes among caregivers and their children, and the healthcare cost and utilization rates among PRx participants.

## 1. Introduction

Food insecurity (FI), defined as a lack of access to enough food for all members of the household [1], is a pervasive global public health concern arising from numerous structural inequities including the accessibility of affordable foods. In the United States, in 2023, 13.5% of households experienced FI, which represents a significant increase over the previous year [1]. The rate of FI was even higher in households with children (17.9%), though this rate was unchanged from the previous year [1]. FI disrupts early childhood nutrition, which is critical for child development and lifelong health [2]. FI is also associated with poor diet quality [3], disordered eating [4], and increased risk of diet-related chronic conditions, including diabetes, hypertension, heart disease, and obesity [5,6,7]. Beyond FI, which primarily focuses on the quantity of food available, nutrition security is a construct that seeks to focus attention on the quality and healthfulness of food available. According to the United States Department of Agriculture, nutrition security is defined as consistent and equitable access to healthy, safe, and affordable foods for optimal health and well-being [8]. Fruits and vegetables contain many essential nutrients, phytochemicals, and fiber that help maintain vital organ functions and are associated with healthy weight maintenance. Strong evidence suggests that an eating pattern that includes approximately five daily servings of fruits and vegetables is associated with reduced mortality [9]. However, many adults and children fall short of this recommendation, particularly those facing FI [3]. Intervention strategies that address FI and nutrition security in households with children may help increase fruit and vegetable consumption and improve the health of children and families [10].

Humanitarian organizations have long been at the forefront of addressing food access and nutrition in vulnerable populations. In recent years, medical and community-based organizations have come together to formally tie access to healthy food with medical interventions in an effort called “Food as Medicine”. In the US, some such strategies including medically tailored meals/groceries, produce prescription interventions (PRxs), federal nutrition programs (FNP), population-level healthy food policies, and nutrition education have been undertaken in adults, but there are fewer pediatric or family-based Food as Medicine programs [11,12]. PRxs, which provide increased access to fruits and vegetables (F/V), are an emerging approach to addressing FI and diet-related risk in children [13,14]. Recent literature suggests that pediatric PRxs are associated with increasing nutrition- and culinary-related knowledge, behaviors, and skills [14,15], weight management [16], increasing F/V consumption [17], and improving FI [18]. Additional studies on the implementation of pediatric-based PRx strategies, views of caretakers, and the impact of diet-related behavior are also emerging.

Pediatric-focused qualitative studies have elicited perspectives from caregivers on voucher-based PRx programs [19], often in partnership with farmers’ markets [20,21,22]. A few have explored caregivers’ attitudes toward prepared bags or boxes of produce [14,23]. These studies have described an overall positive trend around increasing perceived nutritional knowledge, F/V consumption, and FI. However, to our knowledge, there has been limited exploration of families with children and their experiences with home delivery PRx programs.

We previously conducted a mixed-methods study to explore the feasibility and impact of a pediatric-based PRx known as the Family Lifestyle Program’s Produce Prescription Initiative (FLiPRx). FLiPRx offers produce home-delivery services and virtual nutrition education for families with young children (0–5 years old) who live in food-insecure households and are at risk of diet-related chronic illness. The qualitative findings of this feasibility study conducted at the mid-intervention point (after 6 months of the intervention) and quantitative findings collected post-intervention (after 12 months of the intervention) have been reported previously [24]. In brief, the FLiPRx feasibility pilot showed high participant satisfaction in the program and an increase in F/V consumption for some children following the intervention, but no difference in FI scores. The pilot qualitative interviews highlighted caregiver-reported improvements in access to F/V through the intervention.

As a follow-up to our previous feasibility and acceptability pilot and mid-point qualitative study, we conducted a post-intervention qualitative study to explore the lived experiences of families who completed the 12-month FLiPRx initiative. This study aims to add to the current literature on pediatric PRx in a few ways. First, the home delivery of produce is a novel approach to targeting transportation, food quality, and food availability barriers, which are common for families living in under-resourced neighborhoods and can contribute to family FI and poor diet quality. Second, our study population focuses on families with children under 5 years old, an age of rapid development and an opportunity to establish eating patterns that may continue into adolescence and adulthood. We therefore aim to understand the unique challenges and potential impact of a PRx that offers home delivery and virtual nutrition education in families with young children who are experiencing FI.

## 2. Materials and Methods

### 2.1. Study Design and FLiPRx Eligibility

FLiPRx is a community clinical collaboration, based at two outpatient pediatric clinics in Washington, DC. Beginning in November 2020, adults who presented with children aged 0 to 5 years old (reference child) for well-child visits were screened for eligibility. Families who reported FI through the validated 2-question Hunger Vital Sign screener [25] were referred to FLiPRx by their pediatrician. Additional inclusion criteria included reporting one or more additional risk factors that would put the reference child at risk of developing a diet-related chronic disease. These risk factors included having a diagnosis of overweight or obesity in the reference child; demonstrating an abnormal weight gain trajectory in the child; or having a family member, including a sibling, with a diet-related chronic illness such as pre-diabetes, diabetes, and hypertension. Further eligibility criteria have been described in the previous feasibility study [24].

Eligible families provided their informed consent to participate and were enrolled in the FLiPRx intervention conducted between December 2020 and December 2021. The study was approved by the Institutional Review Board of Children’s National Hospital (Pro00014391).

### 2.2. Intervention

Each enrolled family received 8 pounds of “familiar and novel” fresh, locally grown produce, which was pre-selected by the food delivery company based on seasonality and the availability of produce. Families were also given biweekly surveys after each delivery to tailor produce programmatic offerings based on the global interests and preferences of participants. Items were delivered to participant’s homes every other week for 12 months equating to approximately 192 pounds of produce per family. Approximately 24 hours of nutrition education was offered across the year through various modalities. These included live virtual cooking classes led by a registered dietician, biweekly pre-recorded nutrition videos led by pediatricians, video recipes corresponding to the produce being delivered, and recipe cards that were included in the produce boxes. Nutrition videos as well as reminder and invitation messages for the virtual classes were sent to families via text. Additional intervention details have been previously described elsewhere [24].

### 2.3. Quantitative Data Collection and Analysis

At baseline, participants self-reported basic demographics via an online survey, using the REDCap database [26]. Participants were asked to complete satisfaction surveys and attend live virtual education sessions monthly throughout the intervention. There were no specific disenrollment criteria related to monthly survey completion or class attendance. Participation was assessed by self-report virtual class attendance and video views post-intervention. Retention was determined by the number of enrolled families (receiving produce, education, and survey invitations) at the 12-month time points divided by the total number of originally enrolled participants. Demographic and programmatic data on the full cohort were reported in a previously published paper [24]. Univariate descriptive statistics were used to describe the demographic data. Analysis was conducted using SAS 9.4 statistical software (SAS Institute Inc., Cary, NC, USA) [27].

### 2.4. Theoretical Framework and Design of Qualitative Interview Guide

Given the specific aim of understanding family experiences and acceptability of the PRx intervention and its potential influence on nutrition-related eating behaviors and perceived health, the qualitative interview guide was developed with both a Theoretical Framework of Acceptability [28] and Social Cognitive Theory [29] framework in mind. These frameworks have been utilized in the past to assess healthcare-based interventions, including behavioral and dietary interventions like PRxs.

Qualitative interviews were designed to elicit participants’ opinions about their experience with and acceptability of the program and its potential influence on nutrition-related eating behaviors and perceived health. The Theoretical Framework of Acceptability directed the interview guide to include an exploration of how participants felt about the program, their level of satisfaction with the program, how effective they perceived the program to be, how they were able to engage with the various components of the program, and whether there were any barriers to participation in the program. The Social Cognitive Theory framework allowed the interview questions to explore the impact of the program on family food and nutrition behaviors (access, utilization of the delivered items, shopping, cooking, and eating) and factors of the program that were useful for reinforcing continued participation and implementation of healthy eating behaviors. See the interview guide questions in Appendix A.

### 2.5. Qualitative Interview Implementation and Data Collection

Adult caregivers (defined as >18-year-old caretakers for the reference child), who were enrolled in the 12-month intervention were invited to participate in a 30 min individual interview at the conclusion of the 12-month program. Virtual recorded individual interviews were conducted via Zoom (by GC, LF, AP, and KE) with participants within 2 months of completing the 12-month intervention. Adults who completed the interview were provided a $25 gift card for their time. Interviews were conducted utilizing the semi-structured interview guide developed and discussed above. Two interviewers (LF and KE) worked directly with participants during the program and thus were somewhat known by interviewees. The research team underwent structured qualitative interview training prior to conducting interviews. The training included the mitigation of interviewer biases including confirmation, power dynamics, and over-familiarity biases.

### 2.6. Qualitative Data Analysis

A thematic analysis conceptual framework was used to analyze the qualitative data. Thematic analysis is a commonly used theory in qualitative research. It utilizes a systematic 6-step approach of both inductive and deductive iterative qualitative data coding, which is then interpreted into a comprehensive presentation of themes derived from the data [30].

(1) The research team (AP, GC, HM, LF, NK, and KE) started by discussing potential personal biases, in line with qualitative guidelines [31], then read all interview transcripts to become familiar with the content. (2) A codebook was created using a set of descriptive and interpretive codes that reflected the diversity and patterns within the interview transcripts. All three coders (AP, GC, and LF) were also involved in conducting interviews and were aware of the general demographics of the interviewees but were otherwise blinded to the identity of interviewees. Coding was conducted using the online qualitative analysis software, Dedoose Version 8.3.35 (SocioCultural Research Consultants LLC, Los Angeles, CA, USA) [32]. The team met regularly to further refine the codebook, review the coding process, discuss discrepancies, and reach a consensus in coding until all interview transcripts were coded [33]. (3) Overarching preliminary themes and subthemes were derived from coded passages to describe patterned responses or meaning within the data. Themes were considered overarching topics while subthemes were more specific topics that fell under each theme. (4) A thematic map was created to understand the relationship between the preliminary themes and subthemes. (5) Themes and subthemes were refined over time through group discussion until distinctive definitions were produced for each theme and subtheme. (6) A comprehensive and final narrative report was created with supporting quotes from coded transcripts for all themes and subthemes.

## 3. Results

### 3.1. Enrollment, Participation, Retention, and Demographics

Of the 33 families who were referred, the first 25 who met the eligibility criteria consented and were enrolled in the intervention. From these, 18 families remained enrolled at the 12-month time point and were invited to complete the post-intervention qualitative interview. Fifteen adult caretakers completed the interview, meaning the attrition rate was 40% (10/25). Attrition was due to a lack of response to communication (10 participants did not respond to data collection requests) or active withdrawal (previously reported as five participants requested to withdraw [24]). All interviewees were African American and female, and the mean age was 30.2 (±6.4) years old. The median household size was three, including one adult and two children. Forty-seven percent of adult interview participants were unemployed, 47% had a high school diploma, and 47% were making less than $10,000 per year. The majority of families self-reported the use of federal assistance programs. Demographics did not differ by interview status, indicating interviewees were likely not different from those who did not complete an interview in any of the baseline characteristics we reported. See the baseline demographics of the interviewee and non-interviewee sub-groups in Table 1. The baseline demographic information for all 25 participants at baseline has been previously reported [24].

### 3.2. Qualitative Interviews

Three major themes and corresponding sub-themes were identified (Table 2).

#### 3.2.1. Theme 1

Produce delivery partially alleviated financial challenges, contributing to increased produce consumption patterns.

##### A Subtheme 1a: Delivery of Produce Complemented Participation in Federal Nutrition Programs (FNPs)

Participants reported that restrictions in their personal budget and FNP allotments limited their acquisition of food. As described by a participant, “WIC is great, but it’s frustrating because you get $9 for WIC fruits and vegetables and it’s such a limitation on what you can get”. (P9). As a result of limitations in funds, families felt compelled to prioritize certain grocery items over others. One participant described, “I couldn’t buy more healthy food like I wanted to with SNAP (the Supplemental Nutrition Assistance Program) because I had to worry about other things like eggs and milk and poultry”. (P15). Participants reported that the FLiPRx intervention supplemented their food budget by providing fresh produce and allowed families to stretch their FNP benefits or income across the entire month. This led participants to make healthier purchases when they no longer felt restricted by limited financial resources. One participant said, “I wasn’t spending as much money, shopping healthier” (P11) and “It helped me save some of my SNAP money so if we ran out of food, I can go and buy more because I have more stamps left for nutritional foods”. (P15).

Families also described how conditions during the COVID-19 pandemic made it challenging to access and afford healthy foods, as one participant explained, “the cost of food has gotten so high, because of COVID”. But because of FLiPRx, “I really didn’t have to worry about fruits and vegetables, I would just get what I can, even though I’m receiving SNAP”. (P15).

The program served to supplement the families’ budget to benefit families who were already receiving from FNPs to better address gaps in access to healthy food: “I didn’t have to worry about… picking the right stuff or the right quality so having it already picked out and delivered with all the information I needed on how to prepare it, it was probably a lot easier than using the other types of assistance”. (P1).

The delivery aspect was also seen favorably compared to shopping in person at stores as one participant explains: “I didn’t have to go out to the store, you know it’s always a hassle when you take the kids to the store”. (P11).

Even with FNPs, access within their immediate neighborhood was often a challenge: “SNAP is good but I mean by the time you buy meats, and meal makers, a lot of your money is gone… then I went to a lot of places that do online delivery [but] they don’t deliver here. [Grocery store #1] is a great place to go shopping for lower price but quality foods, you can’t pay with SNAP online and have it delivered. You can with [grocery store #2] but [store #2] is expensive”. (P9).

Another participant noted: “I still would have to travel farther to get good quality food. And then, if I did, sometimes the grocery stores that are farther didn’t offer WIC stuff. And then the closer grocery stores will be out of the stuff. So it was a mess. So say one grocery store has one thing and another grocery store has another thing, but both of those things are on the same ticket, so I have to choose which grocery store and what I’m going to leave out”. (P10).

##### B Subtheme 1b: Delivery of Produce Alleviated Stress of Financial Consequences from Food Waste

Families described that prior to the intervention, they felt compelled to prioritize certain grocery items over others due to financial limitations. Families described that this lack of funds created a predilection toward low-cost, shelf-stable foods before the program, “[I] didn’t have the knowledge and I just didn’t know where to get affordable [healthy foods] because the most unhealthy things are the cheapest”. (P13). Another family mentioned, “I buy more shelf-stable things… that I know would be there and would last past a certain amount of time”. (P1).

Families described that the free produce allowed them to try F/V in a low-risk setting without financial repercussions, and this exposure encouraged the adoption of new foods into the typical eating pattern. “[T]here was no better situation for me to use [the produce]. I didn’t have to necessarily pay for them and if we didn’t like them we know to never buy them again or if we did like them, hey we just sampled this food for free, and now we know that we can add it into our repertoire. It allowed me to see what [my son] does like and what he doesn’t without me spending money”. (P9).

This financial freedom allowed caregivers to see what produce their children enjoyed, which motivated them to continue to incorporate these foods into homemade meals. “Because there were a lot of [new] foods that were introduced I saw her liking, I was able to plan and make meals and she enjoyed it”. (P11).

#### 3.2.2. Theme 2

The intervention positively shifted nutrition- and cooking-related knowledge and behavior of families.

##### A Subtheme 2a: Exposure Encouraged Families to Experiment with Produce and Diversify Their Diet

Prior to the program, families reported monotonous eating patterns characterized by a very limited range of foods and the preselected delivery of produce encouraged diversification of their food choices. “When you’re used to eating fast food all the time, and then you get vegetables, you don’t have a choice but to try new things and then you start liking them more. I didn’t even like brussels sprouts… but then I started eating [them] more and more”. (P2) The variety of produce introduced them to unfamiliar F/V, disrupting the monotony of their diets. “Most of the foods that were in the box were new for me and my family, like the beets and yams, red potatoes—my family never ate that before”. (P13). The introduction and repeated exposure to novel produce encouraged families to embrace new F/V, “It definitely did give me the opportunity to at least think about trying other foods; whereas, if I had the choice, I will be less likely to engage in stuff that I’ve never tried before”. (P10).

Another participant admitted being surprised that “some of the [produce] I really didn’t think [my son] would like, but once I made them, he liked them, so if he likes it, I’ll keep making it”. (P9).

In addition, experimentation with unfamiliar produce was also facilitated by the program’s educational materials “I like [the program’s] recipes because it gave me and my daughter the opportunity to get in the kitchen and experiment with different foods together”. (P4).

The delivered bags were received with curiosity from children: “… my kids don’t like [fruits and vegetables]. But as the time went on, we had different things in the bag every time it was like they was curious about what it was and how it will taste”. (P3). The produce bags also appealed to children of different ages, allowing families to provide nutritious items in a developmentally appropriate manner: “my [one year old] baby goes when the [produce] bag is hanging down she gets a fruit out and just eat it versus her grabbing a bag of chips”. The same caregiver describes that for her six year child with autism spectrum disorder, it’s often challenging for him to verbalize what he needs so having a bag of produce present in the house makes it easier “for him to get [produce] and ask ‘can I have this’… so it helped [my kids] to love vegetables”. (P8). Another caregiver reflected on how they have found ways to introduce produce to their toddler: “So we were just using blenders to make smoothies for her and stuff like that. Just giving her veggies and fruits and pushing forward with that”. (P11).

In order to avoid wasting food, participants described learning ways to incorporate unfamiliar food items into their diets. “I’ve looked up the recipe to make beets taste better for us because you guys put beets in there so instead of giving them away I just try to incorporate it and maybe like a food or maybe salad. Sometimes they eat it and sometimes they don’t”. (P12).

##### B Subtheme 2b: Changes in Food Purchasing and Reduction in Processed Food Consumption

Participants described increases in their food literacy, produce resourcefulness, and cooking efficacy. Experiential learning through program involvement and passive exposure to fresh, seasonal produce was described to improve many participants’ food literacy. “We’ve gotten to the point that I know how to buy the fresh things that are in season, when’s a good time to buy them and how to prepare it”. (P1) Another caregiver noted, “I never in the past would have seen myself going into the store purchasing eggplants”. (P4).

Participants felt empowered to increase the quantity and variety of produce they purchased, which in turn reduced the quantity of processed foods they consumed. “Because I’m getting more produce, I don’t buy as much of the snacks and, like the processed stuff that the kids like”. (P1). Caregivers described finding ways to replace processed snacks with homemade alternatives: “We try to stay away from the sugars and processed foods and try to do more [with] the natural vegetables and fruits. Instead of buying a bag of chips, I’ll buy some potatoes and put them in the oven and try to make homemade chips”. (P6).

Additionally, participants’ increased food literacy empowered more confident and informed decision-making, impacting their budgets and taste preferences. “It gave us a wider variety, so instead of just sticking with the traditional fruits and vegetables, it allowed me to look at if that one brand or one type is too expensive, well, I know I can use like the Chinese cabbage and kind of fix it this way and it’ll taste better. So it gave me another variety instead, it spiced up what we eat on a regular basis”. (P9).

This diversification in shopping habits was echoed by others: “we add more variety as far as the produce we purchase and actually use, before it was pretty pretty plain maybe opening like a can of green beans or a can of corn or something like that, but we’ve gotten to the point that I know how to buy the fresh things that are, you know, in season when’s a good time to buy them and things like that, and how to prepare it and how to serve it so that my toddler will likely eat some of it before they eat the other stuff”. (P1). Another family reflected on the changes they have seen in their children: “[Now]…. when I’m in the grocery store, the girls are mainly asking for fruits and vegetables”. (P7).

Some families described improvements in their children’s health status as a result of new eating patterns. “My children had high cholesterol when I joined the program, so it pushed me to start using a different variety of vegetables. Since I’ve been using a different variety of vegetables, during their last physical they no longer suffer from high cholesterol and my daughter is no longer borderline diabetic”. (P4).

##### C Subtheme 2c: Participation Encouraged Family Quality Time

Many participants were creative in trying to engage their families with the produce provided. Families felt encouraged to cook together more and one participant explained, “Me and my daughter would have a cookoff and cook together and see whose soup would be the best”. (P11). This shared family time was shared by others: “I like [the program’s] recipes because it gave me and my daughter the opportunity to get in the kitchen and experiment with different foods together”. (P4). Participants also described how changes in the diet of one family member influenced another. One participant noticed her toddler child began to model the behaviors of the rest of the family stating, “[s]he copies off her siblings so seeing them eating [the produce] or me eating it [then] she wants to eat it [and] to explore her taste buds”. (P8).

The delivered produce bags created a sense of excitement for children and an event to bond over as a family: One caregiver explained, “when the bag comes you know we got to a point where he be like ‘oh mommy my bag’s here can I see what’s inside?’ Sure, go ahead. He liked the bags, he looked forward to them when they came”. (P6). Another added, “we was really enjoying the meal making together and all the different fruits that we haven’t tried that we probably can’t find at the grocery store”. (P7).

Participants also described utilizing different cooking methods to adapt foods to meet their family’s needs and preferences. They expressed excitement regarding passing these skills to their children and future generations of their families. “It’s different ways of cooking because growing up, we always ate greens, but we would boil them. We never put them in the oven or made a salad, so it’s a good thing that I learned how to do that with my children, so they can learn and teach their kids how to do it once they get old enough”. (P15).

#### 3.2.3. Theme 3

Familial and programmatic barriers affected participation and engagement.

##### A Subtheme 3a: Identified Barriers to Participation

Many participants highlighted that time was a significant barrier to utilizing the produce and other aspects of the program. “Well, just with the [COVID-19] pandemic going on and I have four children, so I have a busy schedule and sometimes it was just time”. (P13). Responsibilities including jobs and taking care of young children made it difficult to fully engage with the program’s cooking classes, videos, and preparation of the produce offered. “Well, it was just the schedule, having to do things with my kids at the time and I could not really focus on the classes with the girls being here and I had to tend to the baby, so it was difficult to pay attention like I wanted to”. (P15). Families noted that they enjoyed the classes, but the timing of the classes made it challenging to attend. “The cooking classes I like, but the time of them is normally when we’re settling down in the house and trying to get dinner going so quite a few of them I did not get a chance to make because my house was too hectic for me to be able to focus on it now”. (P9).

While the majority of participants appreciated receiving the different fresh produce, a few participants stated that they were not interested in trying some of the given items. However, participants also noted that exposure through the class encouraged the use of new items. “I was more likely to try something if they cooked it [in the cooking class] but if there was something that they gave [in the bag] and I did not know what it was, I was less likely to try it”. (P10).

Families discarded or gave away produce that they were either not interested in or were not able to use before spoiling. Some families noted giving some items to others saying, “I used most of the produce, sometimes I did not really need the [item], so I gave them to a neighbor and she used them, so that helped them”. (P13).

##### B Subtheme 3b: Suggestions for Future Program Direction

Families offered suggestions for improving aspects of the program based on the barriers they identified throughout the year. Participants had various ideas on how to address the timing concerns. For the cooking classes, participants proposed that “a weekend might be better for most families, because Monday to Friday is hectic”. (P8). Another participant suggested having “more than one time slot for [the cooking classes] so maybe doing two classes, a daytime and then another in the evening” would allow for greater participation. (P1). Families suggested that “the cooking [class] would be better if it was pre-recorded and we could stop, pause, and rewind” in order to account for the distractions they encounter while at home. (P9).

In trying to minimize waste, participants suggested “getting a survey to see what families like more versus wasting things like the beets and mushrooms that I threw away”. (P8). In addition, participants recommended “switching up things in the box so that way we are not getting the same thing every month”. (P12). Families who said that they discarded items because “some of the stuff I just could not eat because I am allergic,” suggested that the program could provide substitutes for those items or suggest alternatives in the recipe cards (P2).

Families also highlighted that they would have preferred more in-person and hands-on aspects of the program saying, “I am more of a hands-on type of person so if COVID was not in the way and there were more segments of the program where it was more family collaborative”. (P4).

## 4. Discussion

This study explored the lived experience of families with young children at risk of FI and diet-related chronic diseases who participated in a novel 12-month home-delivery PRx. Qualitative interview data revealed themes related to the alleviation of food-related financial hardship through participation in FLiPRx; experimentation with produce and changes in food purchasing, which led to an increase in produce consumption and a decrease in processed food; and areas for PRx program improvement. This is consistent with previous qualitative research conducted before [15,21,34] and after the COVID-19 pandemic [19,23], which supports that PRxs may have the potential to support families with young children to improve food quality and quantity through healthy-eating behavior-change support.

Our study adds to the current pediatric PRx literature in a few ways. First, the novel home-delivery system addresses the intersection of financial, transportation, and neighborhood factors, which exacerbate FI. Second, it highlights the ways in which PRx programs can play a role in complementing FNPs to encourage improved dietary quality. Third, it explores the ways in which exposure to produce impacts the palatal development in children.

### 4.1. Financial Hardship and Its Association with FI and Decrease in Nutrition Quality

Within our PRx study population, participants frequently explained that financial limitations led to a lack of variety, decreased nutritional value, and monotony of diets prior to program involvement. Families faced limited household funds, which also consisted of inadequate federal assistance resources, lacking access to full-service grocery stores, and the high cost of food during COVID-19 and resulting economic inflation, which has also been described in other COVID-19-related qualitative research in the US [35] and across the globe [36]. Poverty and financial hardship predispose individuals to FI, reduced diet quality [37], and increased toxic stress [38], which result in an increased risk of diet-related disease in children and adults [7,39,40,41]. Although FNPs are designed to support families with FI and improve access to healthy food [42], their allotment value is often insufficient [43]. While COVID-19 pandemic-related federal policy changes from 2020–2023 had increased the FNP allotments, these policies have since expired. The FLiPRx 1.0 intervention took place during the COVID-19 pandemic and reflects the experiences of many participants who received FNPs. Participants felt FLiPRx increased the availability of produce in the home, enhanced their purchasing power, and reduced cost-based avoidance of F/V. Participants also noted that their diet quality improved through consistent exposure to both familiar and novel produce supplied by the intervention. Similar results were reported by Brown et al. 2022, conducted during the COVID-19 pandemic, where participants picked up boxes of produce at their outpatient clinic sites and found that participation in the program allowed families to stretch their grocery budget to last longer through a month. The caregivers also noted that the exposure to produce has encouraged them to experiment with how to include more produce in their children’s meals and has made produce a routine part of their family’s eating habits [23].

### 4.2. Impact of PRx on Diversification of Childhood Palate and Family Diet

Participants described that the introduction and repeated exposure to produce through PRx participation catalyzed positive changes in the family’s food habits and nutrition literacy. Participants reported embracing novel F/V they received during the program and replacing nutritionally poor and calorically dense foods. Participants reported that the program allowed them to experiment with incorporating new foods, recipes, and cooking modalities into their children’s diets with minimal financial risk. Overall, families felt equipped with new nutrition knowledge and cooking skills to make informed dietary choices and adapt foods to meet their household needs. These findings align with our previously reported quantitative results, showing a significant increase in F/V intake in a sub-group of participants in this cohort [24]. Other studies have found similar results, demonstrating the positive relationship between the amount and variety of F/Vs available in a home and the likelihood of F/V consumption [44,45]. Burrington et al. (2020) studied an online-based PRx where families then picked up self-selected produce from community sites and found in their qualitative interviews that the intervention encouraged increased consumption of produce through more home-cooked meals and family bonding time through cooking [34].

PRxs provide families with both familiar and novel produce without the economic burden. This is critical because while children learn to accept a variety of healthy foods through exposure in their early feeding environment [46], it can require repetition of unfamiliar foods 8 to 15 times before it becomes part of a child’s palate [47]. Higher-income households often have more resources to spend on food and can therefore withstand the cost of uneaten novel food items and are more likely to repeatedly offer foods that their children initially reject, resulting in a so-called term we call a “privilege of waste” [48]. Conversely, as we heard from our program participants, lower-income households minimize the economic risk of food waste by offering foods they are confident their children will accept, which often include calorie-dense, highly processed, nutrient-poor food [19,49,50]. Nationally, food-secure households spend 16% more on food than food-insecure households of the same size [1]. This lack of diversity inadvertently restricts children’s exposure to a small variety of predominantly low-quality and highly processed foods [46,48], which are associated with poorer health outcomes [9,51,52,53]. FLiPRx participants described eating less ultra-processed foods and cooking more meals at home during the intervention. If sustained, this positive behavior change can impact long-term health as family meals are protective against poor dietary intake [54,55,56], excessive weight gain [57,58], and disordered eating behaviors in children and adolescents [59,60,61]. With this lived experience in mind, clinical management of early child feeding should recognize that FI is a healthcare concern that requires medical intervention in families who have unmet healthy food needs. Limited access to affordable healthy food is a serious challenge in managing health, and PRxs are an example of family-centered interventions that support the childhood development of a high-quality diet and improve food and nutrition security, which may present long-term health benefits [48].

### 4.3. Framework of PRx as a Food as Medicine Intervention for FI, Nutrition, and Chronic Disease Risk Management

The results discussed above have led us to develop a proposed theoretical framework for the impact of PRx on food and nutrition security, which is summarized in Figure 1. This framework is supported by the existing literature on FI and FI-related maladaptive coping strategies [45,62], the impact of PRx on FI and nutrition-related behaviors as observed in the current study and elsewhere [24,34,63,64], and qualitative data from clinicians who found PRx integration within the healthcare setting feasible and helpful [65]. This theoretical framework can serve as a guide to future Food as Medicine program development, implementation, and evaluation. Key elements of the framework that should be included in future PRx programs include the “prescription” of produce/food through healthcare partners, home delivery of free or low-cost produce that offers a large degree of food sovereignty and autonomy, experiential nutrition and culinary education, and mixed-methods evaluations of participation and food and nutrition security. Additionally, subsequent to a short-term intervention, there is the need for ongoing engagement with healthcare providers to reassess FI, other social drivers of health, and progression of diet-related chronic disease.

Additionally, the programmatic feedback provided by our participants focused on considerations for the timing and format of nutrition classes as well as increased participant choice over the types of produce delivered. The incorporation of the values, priorities, and voices of participants is critical to delivering a culturally sensitive and patient-centered intervention [66,67]. Participant feedback should guide iterative program improvement and serve to guide the field in identifying the optimal program delivery mechanisms, for example, what works, for whom it works, and why it works. These efforts will support effective program implementation and, ultimately, future funding and healthcare policy considerations. Future PRx research in pediatric populations should focus on studying the long-term impact of PRx on health, healthcare outcomes, quality of life, quality of care, engagement, and total cost of care.

This study has several limitations. There was potential for recall and social desirability bias during individual interviews. We also acknowledge a potential self-selection bias in interviewees, which might make less involved or satisfied participants less likely to interview. Additionally, the attrition rate was 40% and attrition was largely due to a lack of response to communication (10 participants did not respond to data collection requests), as opposed to active withdrawal (previously reported as 5 participants requested to withdraw [24]). As previously mentioned by us and other social needs and health equity researchers, more work is needed to understand the barriers and facilitators of program participation, particularly in those who end participation. These data are notoriously difficult to obtain because people who do not fully participate often do so by default, due to a lack of response to communication, posing a barrier to further data collection efforts. As all participant interviews occurred within 2 months of completing the program, we were unable to evaluate the long-term effects of the program on eating habits. The intervention was conducted during the COVID-19 pandemic during which changes in funding for FNPs, social, and charitable benefits occurred, which may have impacted participants’ purchasing power and perceptions of food hardships in a way we were not able to capture. Another limitation is that this work was conducted in the US and the results might not generalize to settings outside of the US. In general, much of the PRx work has been conducted in US populations and settings, with very little work performed elsewhere [11,68], which is a limitation of the field of Food as Medicine at large.

## 5. Conclusions

Our qualitative findings support the emerging literature suggesting that PRxs improve food security status, diet quality, and financial flexibility among lower-income families with young children at risk of FI. The qualitative results have helped to create the framework discussed above, which can be utilized by healthcare systems to implement and assess their own PRxs. Important aspects of the framework include identifying those with FI in a clinical setting as a medical problem, developing a PRx program as part of the clinical management of FI, providing nutrition education and produce, and assessing the impact on participant diet in the short term and the impact of lifestyle change and health outcomes in the short term. Ongoing feedback and discussion with families, the health system, and implementers are necessary throughout the process to assess the success of the program. Our qualitative results should be followed by quantitative studies on the impact of home-delivery programs for the prevention and treatment of diet-related diseases. Future studies should also examine the sustained impact of PRx on behavioral and health outcomes among caregivers and their children, as well as the healthcare cost and utilization rates among PRx participants. Such research has the potential to influence policies that expand federal PRx funding and lead to the adaptation of PRxs across the healthcare system.

## Figures and Tables

**Figure 1 nutrients-16-04010-f001:**
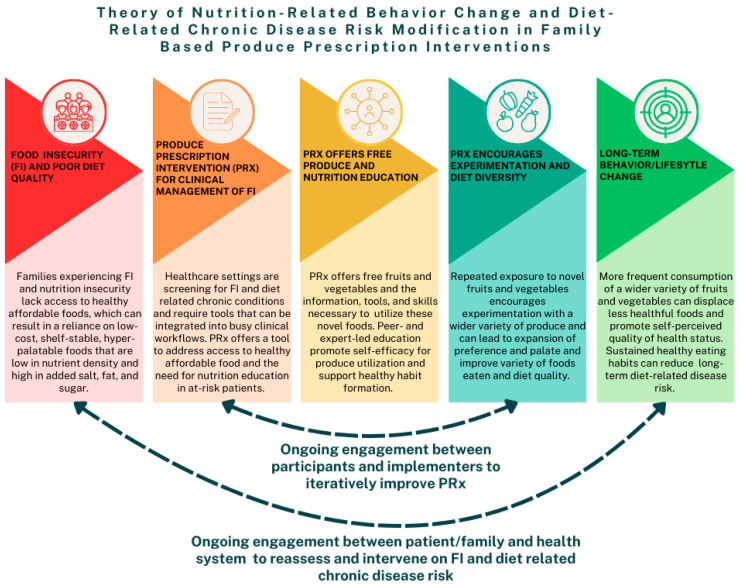
Theory of nutrition-related behavior change and diet-related chronic disease risk modification in family-based produce prescription intervention.

**Table 1 nutrients-16-04010-t001:** Baseline demographic data of parents/caretakers that completed the interview (*n* = 15).

Variable	Description	Response	
		Interviewees (*n* = 15)	Non-Interviewees (*n* = 10)
Gender (*n*, %)	Female	15 (100%)	10 (100%)
Age (mean, sd)	Age in years	30.2 (6.4)	29.6 (5.6)
Reference child age group (*n*, %)	0–1 years	6 (40%)	5 (50%
	>1–5 years	9 (60%)	5 (50%)
Race (*n*, %)	African American	15 (100%)	10 (100%)
Employment status (*n*, %)	Working Full-time	2 (13%)	2 (20%)
	Working Part-time	3 (20%)	3 (30%)
	Going to school or apprenticeship	1 (7%)	1 (10%)
	Unemployed	7 (47%)	3 (30%)
	Prefer not to say	2 (13%)	0 (0%)
Level of education (*n*, %)	Less than High school	2 (13%)	1 (10%)
	High school diploma or GED	7(47%)	5 (50%)
	Some college	3 (20%)	4 (40%)
	College graduate	1 (7%)	0 (0%)
	Prefer not to say	2 (13%)	0 (0%)
Level of income (*n*, %)	Less than $10,000 a year	7 (47%)	3 (30%)
	$10,001–$25,000 a year	1 (7%)	2 (20%)
	$25,001–$50,000 a year	1 (7%)	3 (30%)
	Prefer not to say	6 (40%)	2 (20%)
Marital status (*n*, %)	Never Married/single	10 (67%)	8 (80%)
	Married or unmarried couple	2 (14%)	1 (10%
	Divorced	2 (13%)	0 (0%)
	Prefer not to say	1 (7%)	1 (10%)
Number of Household occupants (median)	Adults	1	1
	Children (age 0–17 in years)	2	3
Federal assistance program participation (*n*, %)	SNAP	9 (60%)	7 (70%)
	WIC	9 (60%)	5 (50%)
	FRPS	5 (33%)	4 (40%)
	TANF	8 (53%)	6 (60%)
	SSI	3 (20%)	3 (30%)

FRPS: Free/Reduced Price School Lunch, GED: General Educational Development, SNAP: Supplemental Nutrition Assistance Program, SSI: Supplemental Security Income, TANF: Temporary Assistance for Needy Families, WIC: Special Supplemental Program for Women, Infants, and Children.

**Table 2 nutrients-16-04010-t002:** Themes, subthemes, and representative quotes.

**Theme 1: Produce delivery partially alleviated financial stress, contributing to increased produce consumption patterns**	
** *Subtheme 1a: Delivery of produce complemented participation in Federal Nutrition Programs (FNPs)* **	*“Getting the bags of fruit and vegetables helped me cut back on getting those type of items with my food stamps and I’ll be able to pick those up for when we don’t have no more of the bag and I could be able to go, like the end of the month, like the 22nd or something until the next month”. (P3)* “[Y]*ou know with getting stuff from you guys it helped me save more of my food stamps because, like I said I didn’t have to worry about vegetables and that’s, the main thing when you want to eat healthy that’s more important than anything else, just you know good vegetables, fresh vegetables…you don’t get anything fresh from these stores really”. (P15)*
** *Subtheme 1b: Delivery of produce alleviated stress of financial consequences from food waste* **	*“Your all program is very beneficial because I don’t have to go to the store and, at the end of the month, or we really don’t run out of food that much because of the you know produce you all give us so it’s a great program”. (P5)* *“I like how the produce itself was delivered and not like the means to get it because things were already picked out for me and they were good quality, so I didn’t have to worry about whether or not I was picking the right stuff or the right quality so having it already picked out and delivered and with all the information I needed”. (P1)*
**Theme 2: Intervention positively shifted nutrition- and cooking-related knowledge and behavior of families**	
** *Subtheme 2a: Exposure encouraged families to experiment with produce and diversify their diet* **	*“The experience has been awesome; it’s introduced me to a lot of different vegetables that I never would have cooked in the past”. (P4)* *“I believe it was getting the different varieties of getting the vegetables in the vegetable and fruit bag cause I really wasn’t a vegetable or fruit person, but when I started receiving that I be eating more vegetables and fruits now”. (P2)* *“I was never really a squash or zucchini person and I tried it for the first time and I actually liked it”. (P5)*
** *Subtheme 2b: Changes in food purchasing and reduction in processed food consumption* **	*“Buying different vegetables as well, because I didn’t buy much squash or beets and things like that. It’s just good that I can now buy that and just put it in my salad or just eat more salads than just fried foods”. (P15)* *“We take more time [in the produce] department… and take our time and pick vegetables and sometimes we would pick something that we wouldn’t pick on a regular”. (P6)* *“[T]he majority of times we probably were either eating fast food or going out. So, we don’t do that often anymore”. (P14)*
** *Subtheme 2c: Participation encouraged family quality time* **	*“We love those* [educational materials] *it actually helps us prep dinner, make dinner. The girls love to help”. (P7)* *“I have a 13 year* [-old child] *and my daughter is about to be two* [years-old]*, so just to have them in the kitchen with me doing different things”. (P15)* *“My daughter she really wanted to help watch the videos and to help cook and make it and stuff like that”. (P3)*
**Theme 3: Familial and programmatic barriers affected participation and engagement**	
** *Subtheme 3a: Identified Barriers to Participation* **	*“Cause I be too busy dealing with my son he has autism, so I’ll be trying to make sure I give him all of the attention that he needs, so me trying to step back and trying to use* [the educational materials] *he’ll start doing a little thing where he want my attention”. (P2)* *“I didn’t really use the FLiP* [recipe] *cards because I couldn’t really do those without help like somebody showing me stuff”. (P3)* *“*[The recipe cards] *would include foods that I don’t eat. Like if it had a meat portion as the protein, I don’t eat meat and then sometimes if I just didn’t have all of the ingredients, I just didn’t use it”. (P10)*
** *Subtheme 3b: Suggestions for Future Program Direction* **	*“I think that it’s great to be able to pick your own food, use your sense of choice and like how I go to YouTube to pick out ingredients for healthy items that I make. I may have that same option by going to pick out those ingredients of my choice versus what was pre-selected in a box for me”. (P13)* *“I think that if the cards had the names of the actual fruits and vegetables on it, I would have been more likely to try, because I could just Google it or use YouTube but because sometimes, I wouldn’t even know the name of [the produce item], I was less likely to try”. (P10)* *“If your child has food allergies, then suggest things that we can substitute in his diet. My son has several nuts, milk, and egg sensitivity since he has G6PD, so we can’t eat things like fava beans. So, if it was something that was geared towards kids that have dietary restrictions, we can better understand like okay you want me to give him all of this, but what if he’s allergic to a lot of it, then what can I supplement?” (P9)*

## Data Availability

The data are not publicly available due to restrictions (privacy and ethical) but are available upon request.

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
