# Peer review of "Qualitative Analysis of a Home-Delivered Produce Prescription Intervention to Improve Food and Nutrition Security"

_nutrients, 2024, doi:10.3390/nu16234010_

Round 1
Reviewer 1 Report
Comments and Suggestions for Authors
Dear authors,
The abstract is very well written, with the authors managing to highlight all the essential parts of the article and all the important elements. The introduction is well-structured and provides an adequate context for the proposed research. It highlights the relevance of food insecurity (FI) as a major public health issue, providing recent statistics and relevant points about its impact on U.S. households, especially those with children. The effects of FI on nutrition, diet quality, and the risks of chronic diseases are mentioned, thus providing a solid foundation for the proposed study. The transition from the general context to the specifics of the study (PRx programs and the FLiPRx intervention) is gradual and logical, creating a coherent narrative thread. However, I suggest the following improvements: It would be helpful to explain the difference between "food insecurity" and "nutrition security" for better understanding. The transition between the sections explaining PRx and FLiPRx could be made smoother to maintain the continuity of the text.
In terms of methodology, the process of selecting participants is well-described, highlighting the eligibility criteria and the use of a validated screening tool (Hunger Vital Sign). The integration of both quantitative and qualitative methods provides a comprehensive perspective on the impact of the intervention. The use of the Theoretical Framework of Acceptability (TFA) and Social Cognitive Theory (SCT) to structure the interviews is a strength, bringing theoretical rigor to the study. A more detailed explanation of the reasons why families might withdraw from the study and the ways to reduce attrition would bring greater clarity to the retention process. Additionally, explaining how the quality and accuracy of data from the bi-weekly surveys were ensured would improve the understanding of the administration process. Considering that some members of the research team (LF and KE) were known to the participants, it would be useful to discuss potential influences on interview responses and how these were managed.
The results are very clearly presented by themes and subthemes and are beautifully represented graphically. Additionally, the results are very well discussed. What needs improvement are the conclusions. They do not sufficiently highlight the obtained results or the implications of the study.
Author Response
- Comment 1: It would be helpful to explain the difference between "food insecurity" and "nutrition security" for better understanding.
- Response 1: We have included definitions of both food insecurity and nutrition security and noted the key difference is a focus on quantity (FI) vs a focus on quality (nutrition security).
- Comment 2: The transition between the sections explaining PRx and FLiPRx could be made smoother to maintain the continuity of the text.
- Response 2: We have made the introduction sentence to the paragraph on FLiPRx easier to read and we hope this smoothes the transition from PRx to FLiPRx.
- Comment 3: A more detailed explanation of the reasons why families might withdraw from the study and the ways to reduce attrition would bring greater clarity to the retention process.
- Response 3: We have included in the discussion the attrition rate and reason for withdrawal (by active request or by default). We have also included statements on the need for further information about how to address attrition.
- Comment 4: Additionally, explaining how the quality and accuracy of data from the bi-weekly surveys were ensured would improve the understanding of the administration process.
- Response 4: Thank you for this comment. We would like to refer to section 2.3 of the methods in which we discuss the surveys as well as the previous paper published in Nutrients (reference #24) which discusses the survey process in detail. This was not included in the current manuscript as our focus was on the qualitative outcomes.
- Comment 5: Considering that some members of the research team (LF and KE) were known to the participants, it would be useful to discuss potential influences on interview responses and how these were managed.
- Response 5: The authors have made the following additions and improvements:
- In the Method section 2.5 Qualitative Interview Implementation and Data Collection , a discussion has been added to specifically address bias mitigation training prior to the interview stage related to the fact that some of the interviewers were known to participants.
- In the Methods section 2.6 Qualitative Data Analysis, have been added to describe that at the analysis stage, the research team were blinded to the identities of the interviewees.
- We would also like to highlight that in the Discussion section, we have reviewed social desirability as a potential limitation of the study design.
- Comment 6: What needs improvement are the conclusions. They do not sufficiently highlight the obtained results or the implications of the study.
- Response 6: The authors have made the following improvements to the conclusion in order to reiterate the major findings of the study and the future implications:
- Conclusion: discussion of the study results and subsequent theoretical framework created
- Conclusion: a discussion has been added around potential policy implications from the study.
Reviewer 2 Report
Comments and Suggestions for Authors
Overall, it is a well-organized and structured research paper; however, it
requires refinement in specific sections for clarity and completeness.
Line 54 : It would be beneficial to provide a brief overview of the health benefits associated with various fruits and vegetables.
Line 157-170: it would be beneficial to make the paragraph more understandable for readers who are not familiar with this methodology.
Additional comments
The presence of numerous abbreviations make it difficult for a reader to fully understand the text and its essence. If it can be simplified in some places, it would be much easier for the reader.
Author Response
- Comment 1: Line 54 : It would be beneficial to provide a brief overview of the health benefits associated with various fruits and vegetables.
- Response 1: The authors have included information on the health benefits of fruits and vegetables: “Food items such as fruits and vegetables contain many essential nutrients, phytochemicals, and fiber that help maintain vital organ functions and a healthy weight and their consumption is associated with reduce mortality”
- Comment 2: Line 157-170: it would be beneficial to make the paragraph more understandable for readers who are not familiar with this methodology.
- Response 2: The paragraphs have been refined by providing a definition for both thematic analysis and Dedoose software. The thematic analysis methodology has now been re-written to represent the six-step process which have now been defined and written in a step wise manner for better clarity.
- Comment 3: The presence of numerous abbreviations make it difficult for a reader to fully understand the text and its essence. If it can be simplified in some places, it would be much easier for the reader
- Response 3: We have removed abbreviations that were used infrequently but kept abbreviations that were use more than 12 times throughout the manuscript (FI - 14x, PRx - 21x, USDA - removed, FAM - removed, FNP - used 12 times, F/V - 14x, FLiPRx - 14x, HVS - removed, TFA - removed, SCT - removed)
Reviewer 3 Report
Comments and Suggestions for Authors
I appreciate the study developed by Caraballo et al. I believe it can be considered for Nutrients after some revisions, as follows:
The abstract is well-structured and gives a good picture of the work conducted by the authors. However, it should be included the main results in a quantitative way.
Lines 38-41: References are missing.
The introductory section should provide a worldwide perspective on food insecurity and provide a better justification for carrying out this study based on that perspective.
More details regarding the qualitative interview design and implementation should be provided.
I suggest you include a flowchart in the Methods section with all the steps you took to conduct your research. A graphical abstract of the manuscript is also recommended.
The Results section is adequate.
Your work would be enhanced by further discussion of similar interventions in other continents. A clear separation of the discussed studies conducted during the COVID-19 pandemic from the others should be pointed out.
Give some practical and policy implications that can be drawn as a consequence of your investigations.
Author Response
- Comment 1: The abstract is well-structured and gives a good picture of the work conducted by the authors. However, it should be included the main results in a quantitative way.
- Response 1: quantitative, baseline demographic data has been added to the results section of the abstract.
- Comment 2: Lines 38-41: References are missing:
- Response 2: Thank you for pointing this out, we have now added a reference accordingly (reference #1).
- Comment 3: The introductory section should provide a worldwide perspective on food insecurity and provide a better justification for carrying out this study based on that perspective.
- Response 3: In the introduction we have mentioned that FI is a global issue. However, we feel that the scope of this work is limited by the uniqueness of the US healthcare system and have chosen to focus on US-based initiatives. This has also been addressed as a limitation of the current work, that it is highly specific to the US and may not generalize to settings outside of the US, and a limitation of the field of Food as Medicine at-large.
- Comment 4: More details regarding the qualitative interview design and implementation should be provided.
- Response 4: The authors acknowledge that dividing the methods section into 2.4 qualitative interview designs and implementation and 2.5 theoretical framework for qualitative data is confusing since the theoretical framework discussion of 2.5 is what explains how the interview questions were designed and 2.4 focuses on how the interviews were implemented. As such, we have rearranged 2.4 and 2.5 so that 2.4 is focused on the interview guide design comes first and then 2.5 focuses on implementation. We have also moved the first part of 2.6 into 2.5 so the the sections read as follows:
- 2.4 Theoretical Framework and Design of Qualitative Interview Guide
- 2.5 Qualitative Interview Implementation and Data Collection
- Qualitative Data Analysis
- Comment 5: I suggest you include a flowchart in the Methods section with all the steps you took to conduct your research. A graphical abstract of the manuscript is also recommended.
- Response 5: We feel that the level of description in the results section is adequate and decline including an additional figure.
- Comment 6: Your work would be enhanced by further discussion of similar interventions in other continents.
- Response 6: To our knowledge there are no similar interventions in children in other continents. There are no healthcare based produce prescription interventions that focus on children outside the US. We have included a reference that address issues related to FI, food availability, access barriers, food cost in contents other than North American, but none were specifically reported from produce prescription interventions.
- Comment 7: A clear separation of the discussed studies conducted during the COVID-19 pandemic from the others should be pointed out.
- Response 7: The authors have made the following improvements:
- Paragraph 1 of discussion: clearly delineated studies before and after the COVID-19 pandemic
- Paragraph 3 of discussion under “Financial hardship and its association with FI and decrease in nutrition quality”, have added that reference 35 is a study specifically designed to study the impact of covid-19 pandemic on family food insecurity.
- Comment 8: Give some practical and policy implications that can be drawn as a consequence of your investigations.
- Response 8: The authors have now added a sentence within the conclusion which discusses how the theoretical framework derived from the study can be utilized to design and assess PRx programs at other institutions. The authors have added a sentence at the conclusion of the study calling for federal policies to implement PRx within federal nutrition assistance programs.